# Differences in Willingness to Undergo *BRCA1/2* Testing and Risk Reducing Surgery among the General Public, Cancer Patients, and Healthcare Professionals: A Large Population-Based Survey

**DOI:** 10.3390/jpm12050818

**Published:** 2022-05-18

**Authors:** Yoon Jung Chang, Seungyeon Cho, Jungnam Joo, Kum Hei Ryu, Sangwon Lee, Juhee Cho, Myong Cheol Lim, So-Youn Jung, Jai Hong Han, Eun Sook Lee, Sun-Young Kong

**Affiliations:** 1Division of Cancer Control & Policy, National Cancer Control Institute, National Cancer Center, Goyang 10408, Korea; eunicemd@ncc.re.kr (Y.J.C.); aprilsycho@gmail.com (S.C.); 2Department of Cancer Control and Policy, National Cancer Center Graduate School of Cancer Science and Policy, Goyang 10408, Korea; mclim@ncc.re.kr; 3Ethical, Legal, and Social Implications Branch, Research Institute, National Cancer Center, Goyang 10408, Korea; 4Office of Biostatistics Research, National Heart, Lung and Blood Institute, National Institutes of Health, Bethesda, MD 20852, USA; jungnam.joo@gmail.com; 5Center for Cancer Prevention & Detection, National Cancer Center Hospital, Goyang 10408, Korea; kumheiryu@ncc.re.kr; 6Cancer Data Center, National Cancer Control Institute, National Cancer Center, Goyang 10408, Korea; 74915@ncc.re.kr; 7Center for Clinical Epidemiology, Samsung Medical Center, Sungkyunkwan University School of Medicine, Seoul 06351, Korea; jh1448.cho@samsung.com; 8Department of Clinical Research Design and Evaluation, SAIHST, Sungkyunkwan University School of Medicine, Seoul 06351, Korea; 9Departments of Health, Behavior and Society and Epidemiology, Johns Hopkins Bloomberg School of Public Health, Baltimore, MD 21205, USA; 10Immuno-Oncology Branch, Division of Rare and Refractory Cancer, Research Institute, National Cancer Center, Goyang 10408, Korea; eslee@ncc.re.kr; 11Center for Gynecologic Cancer, National Cancer Center Hospital, Goyang 10408, Korea; 12Center for Breast Cancer, National Cancer Center Hospital, Goyang 10408, Korea; goje1@ncc.re.kr (S.-Y.J.); 13154@ncc.re.kr (J.H.H.); 13Department of Cancer Biomedical Science, National Cancer Center Graduate School of Cancer Science and Policy, Goyang 10408, Korea; 14Cancer Outcome & Quality Improvement Branch, Research Institute, National Cancer Center, Goyang 10408, Korea; 15Targeted Therapy Branch, Research Institute, National Cancer Center, Goyang 10408, Korea; 16Department of Laboratory Medicine, National Cancer Center Hospital, Goyang 10408, Korea

**Keywords:** *BRCA1*, *BRCA2*, genetic testing, genetic counseling, prophylactic surgical procedure, salpingo-oophorectomy, mastectomy, information dissemination, clinical decision making

## Abstract

We aimed to understand the decision-making process related to the willingness to undergo *BRCA1/2* genetic testing, risk-reducing salpingo-oophorectomy (RRSO), or risk-reducing mastectomy (RRM) among the general public, cancer patients, and healthcare professionals in South Korea. In total, 3444 individuals (1496 from the general public, 1500 cancer patients, 108 clinicians, and 340 researchers) completed a survey addressing genetic testing and related risk management options in a hypothetical scenario. Differences in intent and associated factors for undergoing the above procedures or sharing test results were analyzed. Overall, 67% of participants were willing to undergo *BRCA1/2* testing, with proportions of the general public (58%), cancer patients (70%), clinicians (88%), and researchers (90%). The willingness to undergo RRSO was highest among clinicians (58%), followed by among patients (38%), the general public (33%), and researchers (32%) (*p* < 0.001). Gender, age, education level, and household income were associated with willingness to undergo genetic testing, RRM, and RRSO (*p* < 0.05). The intent for undergo genetic testing, RRM, and RRSO were affected by many factors. Finally, 69% of the general public intended to share information with family, while this percentage was 92%, 91%, and 94% for patients, clinicians, and researchers, respectively (*p* < 0.05). These results highlight the requirement for developing targeted educational materials and counseling strategies for facilitating informed decision making.

## 1. Introduction

Individuals and families with risk of developing hereditary cancer often need to decide their course of action for avoiding oncogenic development, including whether to undergo genetic testing. Further considerations are required if a pathogenic variant is noted, which involve decisions regarding whether to elect risk management options, such as prophylactic surgery, and whether to share the test results with family members. Previous studies have mostly focused on high-risk populations, including patients with breast and ovarian cancer, or *BRCA1/2* mutation carriers in Western countries [1,2,3]. Understanding decision making has been shown to effectively reduce the incidence of cancer, stage at diagnosis, and general and cancer-specific distress [4].

While the prevalence of *BRCA1/2* pathogenic variants in Korea is similar to that in Western countries [5,6], little is known about related decision-making processes in the Korean population [7,8]. The healthcare and legal systems, as well as cultural issues, should be considered for assessing such decision making [9]. Furthermore, understanding the decision-making process of people at relatively low risk of harboring a pathogenic variant is important, as it can help reduce the risk of developing cancer; however, studies examining individuals of different risk levels are limited [10,11]. Therefore, a large-scale study of the Korean population to examine hereditary breast and ovarian cancer risk-related decision making among different risk groups was warranted.

This study aimed to compare the intent to undergo *BRCA1/2* testing, risk-reducing salpingo-oophorectomy (RRSO), and risk-reducing mastectomy (RRM) among the general public, cancer patients, and healthcare professionals in Korea. In addition, the intent of sharing the results of the *BRCA1/2* testing and recommending the test to family members were assessed.

## 2. Materials and Methods

### 2.1. Participants and Procedures

The general public, cancer patients, and healthcare professionals in Korea were surveyed using different sampling methods for each group. The quota sampling method was used to obtain data from the general public. Demographic data, including age, sex, and area of residence, were collected such that the data of our sample population reflected that of the general population in Korea. Cancer patients were recruited to the expected number of each cancer, with patients consent, at the National Cancer Center and Samsung Medical Center. As gastric, colon, lung, breast, and gynecological cancers are predominant in Korea, samples of 300 patients were obtained for each major cancer subgroup. Data from healthcare professionals, including clinicians, nurses, professors, and researchers, were obtained at the 21st annual fall symposium of the Korean Cancer Association, one of the largest oncological symposiums in Korea, held on 18 November 2016. Individuals unable to communicate in Korean were excluded.

Survey respondents were excluded if data regarding key variables of interest (age, gender, income, education, willingness to undergo *BRCA 1/2* testing, RRSO, or RRM, or willingness to share test results with family members or recommend *BRCA* testing to family members) were missing. For all respondents other than cancer patients, data were excluded if the individual reported a previous diagnosis of cancer in the past year. For researchers, data were excluded if they did not disclose their specialty.

Survey data were obtained from November 2016 to February 2017. Informed consent was obtained from all participants. All members of the general public and cancer patients were surveyed using face-to-face interviews conducted by trained interviewers to avoid misunderstandings regarding the concepts or terminology related to genetic testing. Healthcare professionals were surveyed via self-administered questionnaires. The institutional review boards of the National Cancer Center and Samsung Medical Center approved the study protocol (IRB #NCC 2016-0256 and #SMC 2016-12-040-001).

### 2.2. Measures

A literature review and consultations with healthcare experts in the field were conducted prior to designing the survey, which contained 69 questions addressing the following domains: socio-demographic characteristics (14 items), big data (18 items), precision medicine (12 items), and genetic testing (25 items). The genetic testing section included assessments of willingness to undergo testing, RRM, and RRSO to reduce risk, followed by a hypothetical scenario (Appendix A) in which individuals developed breast cancer at the age of 37 years. For evaluating attitudes toward *BRCA* genetic testing, respondents were asked about three hypothetical scenarios related to *BRCA* genetic testing: (1) intention to undergo *BRCA* genetic testing; (2) intention to undergo RRSO or RRM; (3) intention of sharing genetic testing information with family members. Willingness to communicate test results with family members was also assessed.

### 2.3. BRCA1/2 Testing, RRM, and RRSO Intent

Respondents were informed with simple information about hereditary breast and ovarian cancer risk compared to control and *BRCA1/2*-positive testing, and asked whether they would undergo *BRCA1/2* testing. The response options were “yes” and “no”. The participants were provided a list of reasons regarding their intent, or lack thereof, for testing. These statements were generated from previous studies examining the barriers and facilitators of genetic testing [1,2]. After reviewing the hypothetical scenario related to the harboring of a *BRCA1/2* pathogenic variant, the respondents received information regarding breast and ovarian cancer and the advantages and disadvantages of RRSO and RRM. Subsequent intent for RRSO and RRM and their underlying reasoning were then assessed.

### 2.4. Familial Communication Intent

To determine the intent to share test results and recommend testing to family members, respondents were asked to review a hypothetical scenario wherein they harbored a *BRCA1/2* pathogenic variant and had the following family members: parents, siblings (brother and sister), spouse, and both adult and underage children. Respondents answered the following questions: “Would you share that you have a *BRCA1/2* mutation with family members?” and “Would you recommend that your family members undergo *BRCA1/2* testing?”. Both questions were followed by the multiple-choice question, “With whom/to whom would you recommend?”. The response options included parents, siblings, spouse, adult children, underage children, and friends.

### 2.5. Statistical Analyses

Descriptive statistics were used to summarize the general characteristics of the respondents intent to undergo *BRCA1/2* testing, RRSO, RRM, and familial communication. Chi-squared tests and one-way analysis of variance (ANOVA) with Scheffe’s post hoc tests were performed to compare socio-demographic differences among the study groups. Multivariate analysis involved logistic regression models to obtain odds ratios (ORs) and 95% confidence intervals (CI). All analyses were performed using STATA version 13.1 (StataCorp LLC, College Station, TX, USA); a two-sided value of *p* < 0.05 indicated statistical significance.

## 3. Results

### 3.1. Demographic Characteristics

In total, 3444 individuals (1496 from the general public, 1500 cancer patients, 108 clinicians, and 340 researchers) completed the questionnaire on genetic testing and related risk management options according to a hypothetical scenario. The demographic characteristics are presented in Table 1.

### 3.2. Intent to Undergo BRCA1/2 Testing, RRSO, and RRM

Overall, 67% (*n* = 2322) of participants expressed willingness to undergo *BRCA1/2* testing after being informed about hereditary breast and ovarian cancer risk and *BRCA1/2* testing. In total, 58% (*n* = 867) of the general public were willing to undergo the test, while the percentage was 70% (*n* = 1054) for cancer patients, 88% (*n* = 95) for clinicians, and 90% (*n* = 306) for researchers (*p* < 0.001). Additionally, 36% of participants (*n* = 1234; public, 33%; patients, 38%; clinicians, 58%; researchers, 32%; *p* < 0.001) expressed intent to undergo RRSO, and 27% (*n* = 913; public, 27%; patients, 25%; clinicians 34%; researchers, 28%; *p* = 0.208) expressed intent to undergo RRM.

### 3.3. Decision-Making Predictors for BRCA1/2 Testing, RRSO, and RRM

Table 2, Table 3, Table 4, Table 5, Appendix A show the factors associated with intent to undergo *BRCA1/2* testing, RRSO, and RRM. In a multivariate analysis, respondent type, gender, and monthly household income were significantly associated with *BRCA1/2* testing intent. Respondent type, gender, age, and education level were associated with RRSO intent, and gender and education level were significant predictors of RRM intent. Male subjects from the general public and male cancer patients represented less willingness to undergo genetic testing, RRSO, and RRM compared to females, and there was an obvious difference when we compared these subjects with clinician and researchers.

### 3.4. Barriers and Facilitators of BRCA1/2 Testing, RRSO, and RRM

Figure 1, Figure 2 and Figure 3 depict the differences in barriers and facilitators of intent to undergo *BRCA1/2* testing, RRSO, and RRM, respectively, among the general public, cancer patients, clinicians, and researchers. Participants who intended to undergo *BRCA1/2* testing reported the following reasons: “[the wish] to identify the risk of other diseases” (49%), “[it] might be helpful for cancer treatment” (29%), and “to inform family members of any heritability” (22%). “Burden of cost” (33%), “feeling no need since already diagnosed with breast cancer” (25%), “fear of discovering a genetic mutation” (23%), and “do not trust genetic test results” (16%) were reported as the main barriers among individuals who did not intend to undergo *BRCA1/2* testing (Figure 1).

Factors contributing to RRSO intent included “being at high risk of ovarian cancer” (72%), “commonness of ovarian cancer” (21%), and “family history of ovarian cancer” (6%). Among the barriers reported by those not intending to undergo RRSO, “the fact that I do not have breast cancer yet” (60%) was the most common (Figure 2).

Reasons reported by those expressing the intent to undergo RRM included “test results indicated being at high risk of developing breast cancer” (67%), “because breast cancer is common nowadays” (28%), and “family history of breast cancer” (5%). “The fact that I do not have breast cancer yet” (66%) was the most frequently reported reason among individuals not willing to undergo RRM (Figure 3).

### 3.5. Familial Communication

Figure 4 shows the percentages of participants that expressed the intent to communicate test results and recommend testing to family members if they themselves harbored a *BRCA1/2* mutation. Overall, 82% (*n* = 2830) of individuals intended to share their test results, and 79% (*n* = 2720) intended to recommend testing to at least one family member. Significant differences among groups were observed for both parameters (*p* < 0.05). Intent to share results with family members was reported by 69% of the general public, and by 92%, 91%, and 94% of patients, clinicians, and researchers, respectively. Finally, 71% of the general public, 84% of patients, 92% of clinicians, and 91% of researchers reported intent to recommend testing to family members. Respondents were most likely to communicate test results to their spouses (85%) (Figure 4). Significant differences were observed among the groups (*p* < 0.001); cancer patients expressed lower intent to share results with parents than did members of the general public (19% vs. 39%, respectively) and higher intent to share results with adult children (74% vs. 34%, respectively). Respondents indicated the highest intent to recommend testing to adult children (73%), followed by sisters (66%), parents (35%), brothers (35%), and underage children (31%). A significant difference in intent was observed among the general public and cancer patients (*p* < 0.001); patients were less likely than the general public to recommend BRCA testing to parents (20% vs. 22%, respectively); however, they were more likely to recommend testing to adult children (86% vs. 58%, respectively).

## 4. Discussion

This is the first large population-based study in Asia that comprehensively examined the intent for *BRCA1/2* testing, RRSO, RRM, and familial communication of testing results. The intent to undergo *BRCA1/2* testing and RRSO and RRM procedures reported here is generally comparable with previous rates from systematic reviews, which were 66%, 34%, and 24%, respectively [1,2]. However, the rates of intent to communicate *BRCA1/2* test results with family members in our study are considerably lower than those reported in the US and other Western countries, in which 91–100% of participants reported the intent to communicate with at least one blood relative [3,12,13].

Disclosure of genetic information is influenced by social, cultural, religious, and familial factors [9] and differs between Asian and Caucasian populations [14]; however, few studies have investigated these factors in Asia. These findings indicate a critical need for the examination of cultural and ethnic influences on familial communication of genetic testing results. Several patterns of communication observed in our study are consistent with those reported previously, including more communication with female relatives [13,15], less communication with parents among those older than 40 years [3], and more communication with adult children than with underage children [3].

This study also identified differences in intent and the influencing factors between different risk groups: the general public, cancer patients, and healthcare professionals. As observed in previous studies, cognitive and psychological factors were identified as barriers and facilitators of such intentions [16,17]. Women perceiving high risk of developing breast and ovarian cancer are more willing to undergo *BRCA1/2* testing and prophylactic surgery [18,19]. However, studies have shown that individuals do not always have an accurate perception of their own cancer risk [19]. In the absence of personal experience, an optimistic bias of one’s own risk of developing cancer is common [20]. The present study indicates that a significant proportion of the general public and cancer patients did not have an accurate perception of their own risks of developing breast or ovarian cancer, and high optimism was observed among the general public. In many cases, misconceptions are barriers to genetic testing [21]. Education to counteract these misconceptions is a potentially important step toward facilitating testing and follow-up care. The development of educational materials and the implementation of personalized counseling strategies that provide better information will possibly be an important component of the education process.

This study also investigated the financial concerns and their relationship with intent to undergo genetic testing for breast cancer risk. All participants expressed that cost concerns were the major reason responsible for unwillingness to undergo testing and surgeries. The highest levels of concern were expressed by cancer patients. Consistent with the results of previous studies, higher income was associated with willingness to undergo testing, RRM, and RRSO among cancer patients [22,23,24]. This was indicative of the economic burden resulting from the medical care of patients, also referred to as “financial toxicity” [25,26]. A recent nationwide study in Korea reported that 48% of cancer patients felt burdened by the cost of care [27]. In March 2018, the National Health Insurance System of Korea expanded coverage using a next-generation sequencing panel-based test to reduce the cost of genetic testing by 50%. This rendered genetic testing more affordable, with a potential impact on the willingness of patients to undergo testing [28], especially among those of low socioeconomic background and with a high financial burden. While a previous study has reported the impact of changes in the health insurance system on intent to undergo RRSO and RRM [29], more studies are required to assess the effect of changes in genetic testing policy on the willingness to undergo these procedures. Moreover, the national income level will also be considered as a factor in the attitude about the decision-making process [30].

Our study has several limitations. First, this study examined hypothetical risk rates. It is well known that expressions of willingness do not reflect actual participation rates. This is attributable to several factors, including the availability of testing and preventive measures, as well as the cost and other real or perceived barriers to accessing testing and preventive measures. Levels of willingness to undergo *BRCA1/2* testing and preventive measures for breast and ovarian cancer risk, however, provide some useful indicators of public awareness regarding testing and the potential demand for information about testing. Second, cancer patients were recruited from only two hospitals in Korea. These are two of the five largest cancer hospitals nationwide, accounting for nearly 30% of all cancer patients in Korea. Moreover, the areas of residence of the recruited patients were evenly distributed throughout the country, analogous to the areas of residence of the national population.

## 5. Conclusions

In conclusion, this study expands our current knowledge of hereditary breast and ovarian cancer risk management decisions in an Asian population. The results revealed potential influences on decision making related to *BRCA1/2* genetic testing, RRSO, and RRM procedures, and familial communication among diverse risk groups. The findings of this study provide important indications for the development of targeted educational materials and counseling strategies for individuals in general, as well as for those with breast and ovarian cancer, which will facilitate informed decision making.

## Figures and Tables

**Figure 1 jpm-12-00818-f001:**
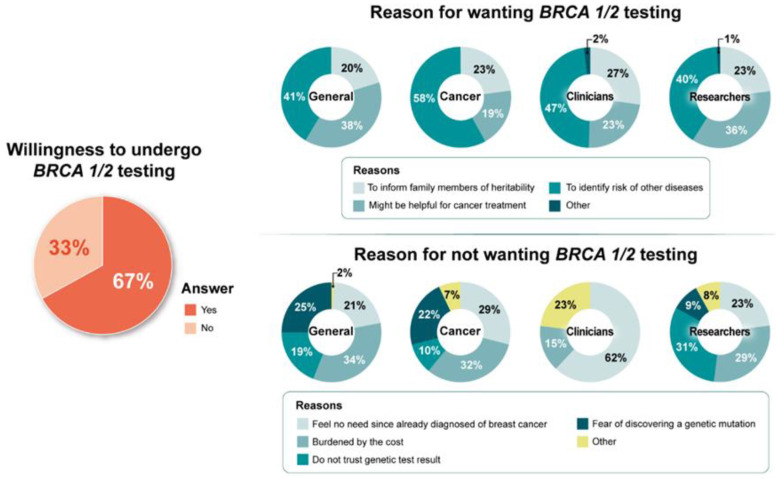
Intent to undergo *BRCA1/2* testing and its reasons by respondent type.

**Figure 2 jpm-12-00818-f002:**
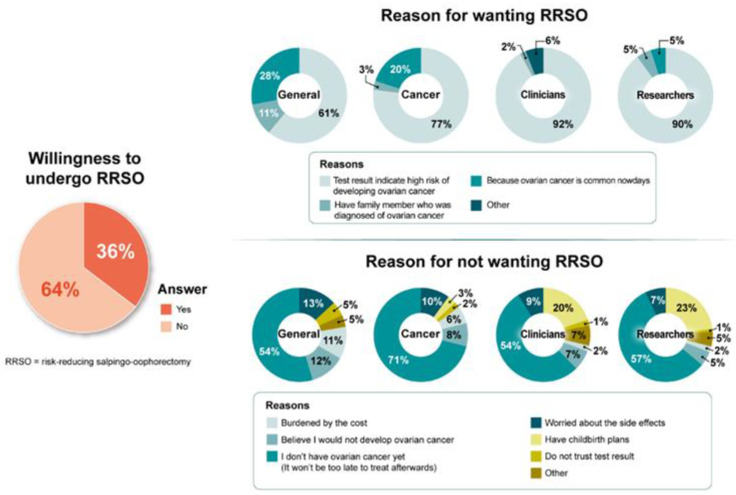
Intent to undergo risk-reducing salpingo-oophorectomy and its reasons by respondent type.

**Figure 3 jpm-12-00818-f003:**
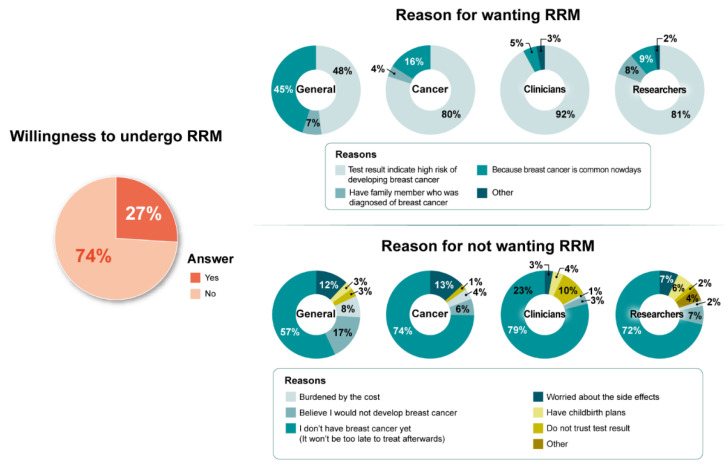
Intent to undergo risk-reducing mastectomy (RRM) and its reasons by respondent type.

**Figure 4 jpm-12-00818-f004:**
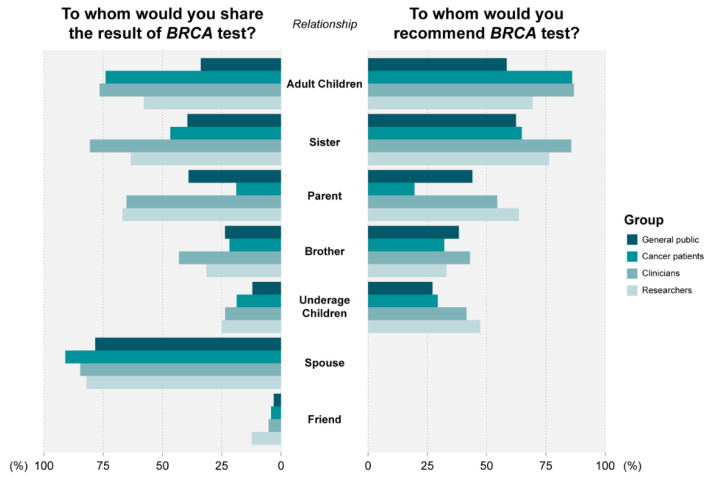
Intent to share *BRCA 1/2* test results with family members and recommend testing by respondent type.

**Table 1 jpm-12-00818-t001:** Sociodemographic characteristics by respondent type.

Variables	All	General Public	Cancer Patients	Clinicians	Researchers
	(*n* = 3444)	(*n* = 1496)	(*n* = 1500)	(*n* = 108)	(*n* = 340)
	N (%)	N (%)	N (%)	N (%)	N (%)
Gender					
Female	1908 (55.4)	723 (48.3)	901 (60.1)	52 (48.1)	232 (68.2)
Male	1536 (44.6)	773 (51.7)	599 (39.9)	56 (51.9)	108 (31.8)
Age, years					
Mean (SD)	47.7 (13.5)	43.0 (12.0)	56.0 (11.1)	40.7 (8.4)	34.2 (7.9)
20–39	1033 (30.0)	615 (41.1)	109 (7.3)	50 (46.3)	259 (76.2)
40–49	796 (23.1)	381 (25.5)	313 (20.9)	40 (37.0)	62 (18.2)
≥50	1615 (46.9)	500 (33.4)	1078 (71.9)	18 (16.7)	19 (5.6)
Education Level					
≤High school	1721 (50.0)	797 (53.3)	924 (61.6)	0 (0)	0 (0)
College graduate	1363 (39.6)	687 (45.9)	522 (34.8)	15 (13.9)	139 (40.9)
Graduate degree	360 (10.4)	12 (0.8)	54 (3.6)	93 (86.1)	201 (59.1)
Household monthly income					
<USD 3000	988 (28.7)	261 (17.4)	668 (44.5)	0 (0)	59 (17.4)
USD 3000–4999	1961 (56.9)	1116 (74.6)	708 (47.2)	18 (16.7)	119 (35.0)
≥USD 5000	495 (14.4)	119 (8.0)	124 (8.3)	90 (83.3)	162 (47.6)
Living area					
Metro	1489 (43.2)	690 (46.1)	516 (34.4)	78 (72.2)	205 (60.3)
Non-metro	1955 (56.8)	806 (53.9)	984 (65.6)	30 (27.8)	135 (39.7)
Perceived health status					
Excellent/Good	1958 (56.9)	1024 (68.5)	665 (44.3)	73 (67.6)	196 (57.7)
Fair/Poor/Very Poor	1486 (43.1)	472 (31.5)	835 (55.7)	35 (32.4)	144 (42.3)

**Table 2 jpm-12-00818-t002:** Univariate logistic regression analysis of variables associated with intent to undergo *BRCA1/2* testing.

Variables	All	General Public	Cancer Patients	Clinicians	Researchers
	(*n* = 3444)	(*n* = 1496)	(*n* = 1500)	(*n* = 108)	(*n* = 340)
	N (%)	N (%)	N (%)	N (%)	N (%)
Group	*p < 0.001*				
General public	1				
Cancer patients	**1.71 (1.47–1.99)**				
Clinicians	**5.30 (2.94–9.55)**				
Researchers	**6.53 (4.52–9.44)**				
Gender	*p < 0.001*	*p = 0.001*	*p < 0.001*	*p = 0.661*	*p = 0.265*
Female	1	1	1	1	1
Male	**0.63 (0.55–0.73)**	**0.71 (0.58–0.87)**	**0.61 (0.49–0.76)**	1.30 (0.41–4.15)	1.58 (0.69–3.61)
Age, years	*p < 0.001*	*p = 0.706*	*p < 0.001*	*p = 0.183*	*p = 0.557*
20–39	1	1	1	1	1
40–49	0.91 (0.74–1.11)	0.90 (0.70–1.17)	0.63 (0.35–1.12)	0.30 (0.07–1.25)	1.76 (0.59–5.21)
≥50	**0.73 (0.62–0.87)**	0.93 (0.73–1.18)	**0.37 (0.22–0.63)**	0.32 (0.06–1.75)	1.03 (0.23–4.70)
Educational Level	*p < 0.001*	*p = 0.166*	*p < 0.001*	*p = 0.461*	*p = 0.026*
≤High school	1	1	1		
College graduate	**1.28 (1.10–1.48)**	0.99 (0.81–1.22)	**1.63 (1.28–2.08)**	1	1
Graduate degree	**4.18 (3.01–5.79)**	3.64 (0.79–16.74)	1.08 (0.60–1.95)	0.48 (0.06–4.01)	**2.24 (1.09–4.61)**
Household monthly income	*p < 0.001*	*p = 0.187*	*p < 0.001*		*p = 0.163*
<USD 3000	1	1	1		1
USD 3000–4999	1.15 (0.98–1.34)	0.94 (0.71–1.23)	**1.82 (1.44–2.30)**	1	2.49 (0.95–6.52)
≥USD 5000	**2.32 (1.80–3.00)**	1.34 (0.86–2.11)	**1.81 (1.17–2.82)**		2.00 (0.84–4.74)
Living area	*p = 0.308*	*p = 0.060*	*p = 0.851*	*p = 0.263*	*p = 0.089*
Metro	1	1	1	1	1
Non-metro	1.08 (0.93–1.24)	1.22 (0.99–1.50)	1.02 (0.81–1.29)	2.30 (0.48–11.05)	1.94 (0.88–4.31)
Perceived health status	*p = 0.458*	*p = 0.160*	*p = 0.024*	*p = 0.024*	*p = 0.095*
Excellent/Good	1	1	1	1	1
Fair/Poor/Very poor	1.06 (0.91–1.22)	1.17 (0.94–1.46)	**0.77 (0.62–0.97)**	6.69 (0.83–53.68)	0.55 (0.27–1.11)

Univariate comparisons were performed using χ^2^ tests and crude odds ratio (95% confidence interval). Boldface indicates statistical significance (*p* < 0.05); *p* values were tested using χ^2^ tests.

**Table 3 jpm-12-00818-t003:** Univariate logistic regression analysis of variables associated with intent to undergo risk- reducing salpingo-oophorectomy.

Variables	All	General Public	Cancer Patients	Clinicians	Researchers
	(*n* = 3444)	(*n* = 1496)	(*n* = 1500)	(*n* = 108)	(*n* = 340)
	N (%)	N (%)	N (%)	N (%)	N (%)
Group	*p < 0.001*				
General public	1				
Cancer patients	**1.25 (1.08–1.46)**				
Clinicians	**2.87 (1.93–4.26)**				
Researchers	0.98 (0.76–1.26)				
Gender	*p < 0.001*	*p < 0.001*	*p < 0.001*	*p = 0.794*	*p = 0.003*
Female	1	1	1	1	1
Male	**0.68 (0.59–0.79)**	**0.56 (0.45–0.69)**	**0.64 (0.52–0.80)**	0.90 (0.42–1.94)	**2.07 (1.28–3.33)**
Age, years	*p = 0.009*	*p = 0.222*	*p = 0.250*	*p = 0.671*	*p = 0.062*
20–39	1	1	1	1	1
40–49	**1.28 (1.06–1.56)**	1.01 (0.77–1.33)	1.46 (0.92–2.32)	1.21 (0.52–2.83)	**1.89 (1.07–3.34)**
≥50	**1.26 (1.07–1.49)**	1.23 (0.96–1.58)	1.28 (0.84–1.95)	0.72 (0.25–2.13)	1.78 (0.69–4.61)
Education Level	*p = 0.030*	*p = 0.028*	*p = 0.685*	*p = 0.888*	*p = 0.819*
≤High school	1	1	1	1	
College graduate	1.06 (0.91–1.23)	1.14 (0.91–1.41)	1.06 (0.85–1.32)	0.92 (0.30–2.81)	1
Graduate degree	**1.37 (1.09–1.73)**	**4.40 (1.31–14.75)**	1.25 (0.71–2.17)		1.06 (0.66–1.68)
Household monthly income	*p = 0.026*	*p = 0.409*	*p = 0.056*	*p = 0.793*	*p = 0.676*
<USD 3000	1	1	1		1
USD 3000–4999	1.03 (0.88–1.22)	1.19 (0.89–1.60)	1.09 (0.88–1.36)	1	0.76 (0.39–1.46)
≥USD 5000	1.34 (1.07–1.67)	1.30 (0.82–2.07)	**1.61 (1.09–2.37)**	0.87 (0.31–2.45)	0.77 (0.41–1.44)
Living area	*p = 0.971*	*p = 0.346*	*p = 0.114*	*p = 0.828*	*p = 0.382*
Metro	1	1	1	1	1
Non-metro	1.00 (0.87–1.15)	0.90 (0.73–1.12)	1.19 (0.96–1.49)	0.91 (0.39–2.13)	0.81 (0.51–1.30)
Perceived health status	*p = 0.161*	*p = 0.075*	*p = 0.94*	*p = 0.066*	*p = 0.741*
Excellent/Good	1	1	1	1	1
Fair/Poor/Very Poor	1.11 (0.96–1.27)	1.23 (0.98–1.55)	1.01 (0.82–1.24)	0.47 (0.21–1.06)	1.08 (0.68–1.71)

Univariate comparisons were performed using χ^2^ tests and crude odds ratio (95% confidence interval). Boldface indicates statistical significance (*p* < 0.05); *p* values were tested using χ^2^ tests.

**Table 4 jpm-12-00818-t004:** Multivariate logistic regression analysis of variables associated with intent to undergo risk- reducing salpingo-oophorectomy.

Variables	All	General Public	Cancer Patients	Clinicians	Researchers
	(*n* = 3444)	(*n* = 1496)	(*n* = 1500)	(*n* = 108)	(*n* = 340)
	N (%)	N (%)	N (%)	N (%)	N (%)
Group					
General public	1				
Cancer patients	1.09 (0.92–1.30)				
Clinicians	**2.32 (1.40–3.84)**				
Researchers	0.83 (0.59–1.17)				
Gender					
Female	1	1	1		1
Male	**0.66 (0.57–0.76)**	**0.54 (0.43–0.67)**	**0.66 (0.53–0.82)**		**1.85 (1.12–3.04)**
Age, years					
20–39	1				1
40–49	**1.24 (1.01–1.53)**				1.55 (0.85–2.81)
≥50	**1.39 (1.13–1.71)**				1.56 (0.59–4.10)
Educational Level					
≤High school	1	1			
College graduate	**1.19 (1.01–1.41)**	1.24 (0.99–1.54)			
Graduate degree	1.32 (0.92–1.88)	**5.19 (1.53–17.63)**			
Household monthly income					
<USD 3000	1		1		
USD 3000–4999	1.05 (0.88–1.25)		1.05 (0.84–1.30)		
≥USD 5000	1.16 (0.89–1.51)		**1.49 (1.01–2.20)**		
Living area					
Metro					
Non-metro					
Perceived health status					
Excellent/Good					
Fair/Poor/Very Poor					

Adjusted OR (95% CI). OR = Odds ratio; 95% CI = 95% confidence interval; multivariate adjustment comparisons were performed using logistic regression (backward procedure, Wald test). Boldface indicates statistical significance (*p* < 0.05).

**Table 5 jpm-12-00818-t005:** Univariate logistic regression analysis of variables associated with intent to undergo risk-reducing mastectomy.

Variables	All	General Public	Cancer Patients	Clinicians	Researchers
	(*n* = 3444)	(*n* = 1496)	(*n* = 1500)	(*n* = 108)	(*n* = 340)
	N (%)	N (%)	N (%)	N (%)	N (%)
Group	*p = 0.224*				
General public	1				
Cancer patients	0.94 (0.80–1.11)				
Clinicians	1.43 (0.95–2.17)				
Researchers	1.07 (0.92–1.39)				
Gender	*p < 0.001*	*p < 0.001*	*p = 0.059*	*p = 0.940*	*p = 0.006*
Female	1	1	1	1	1
Male	**0.76 (0.65–0.89)**	**0.57 (0.45–0.72)**	0.79 (0.62–1.01)	0.97 (0.44–2.15)	**2.02 (1.23–3.30)**
Age, years	*p = 0.660*	*p = 0.353*	*p = 0.397*	*p = 0.126*	*p = 0.315*
20–39	1	1	1	1	1
40–49	1.09 (0.88–1.34)	1.02 (0.76–1.37)	1.27 (0.77–2.10)	0.66 (0.28–1.58)	1.47 (0.81–2.66)
≥50	1.00 (0.84–1.19)	1.20 (0.92–1.57)	1.05 (0.66–1.67)	0.28 (0.07–1.08)	1.67 (0.63–4.42)
Educational Level	*p = 0.019*	*p = 0.097*	*p = 0.177*	*p = 0.284*	*p = 0.596*
≤High school	1	1	1	1	
College graduate	**1.23 (1.05–1.45)**	1.17 (0.93–1.47)	1.25 (0.98–1.60)	0.54 (0.18–1.64)	1
Graduate degree	1.28 (0.99–1.64)	2.99 (0.95–9.36)	1.23 (0.67–2.28)		0.88 (0.54–1.42)
Household monthly income	*p = 0.040*	*p = 0.177*	*p = 0.013*	*p = 0.042*	*p = 0.366*
≤USD 3000	1	1	1		1
USD 3000–4999	1.19 (1.00–1.42)	1.35 (0.98–1.85)	1.13 (0.88–1.44)	1	0.64 (0.33–1.25)
≥USD 5000	**1.34 (1.05–1.71)**	1.23 (0.75–2.04)	**1.87 (1.24–2.81)**	**0.34 (0.12–0.96)**	0.65 (0.35–1.24)
Living area	*p = 0.234*	*p = 0.129*	*p = 0.354*	*p = 0.296*	*p = 0.5*
Metro	1	1	1	1	1
Non-metro	0.91 (0.78–1.06)	0.84 (0.67–1.05)	1.12 (0.88–1.44)	0.61 (0.24–1.56)	0.85 (0.52–1.38)
Perceived health status	*p = 0.530*				
Excellent/Good	1				
Fair/Poor/Very Poor	1.05 (0.90–1.22)				

Univariate comparisons were performed using χ^2^ tests and crude odds ratio (95% confidence interval). Boldface indicates statistical significance (*p* < 0.05); *p* values were tested using χ^2^ tests.

## Data Availability

Available upon request.

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
