# Peer review of "Differences in Willingness to Undergo BRCA1/2 Testing and Risk Reducing Surgery among the General Public, Cancer Patients, and Healthcare Professionals: A Large Population-Based Survey"

_jpm, 2022, doi:10.3390/jpm12050818_

Round 1

Reviewer 1 Report

This is an excellent article, very well structured and written. I highly recommend this article for publication. My observations are minor
Page 2 line 79-80 reference the sampling method of the National Center and Samsung Medical Center

Table 1,2,3,4 y 5
the title of the columns of the tables place them on one line and the n= on another line since everything is piled up

Figure 2 I
n the title of figure 2 there is a missing space between the words risk-reducing salpingo-oophorectomy Figure 3 In the title of figure 3 there is one more space between the word risk-reducing I think they need to put perspectives in the conclusions

Author Response

Thank you for valuable comments. We changed several points according to reviewer's comment. 

  1. Page 2 line 79-80 reference the sampling method of the National Center and Samsung Medical Center
    Response) We changed manuscript for explanation of sampling methods of patients. We recruited patients as quota of each included cancer and the only limitation for inclusion was patient’s consents. We added this points in the manuscript.
  2. Table 1,2,3,4 y 5 the title of the columns of the tables place them on one line and the n= on another line since everything is piled up
    Response) We changed table file as following of reviewers comment.
  3. Figure 2 In the title of figure 2 there is a missing space between the words risk-reducing salpingo-oophorectomy  Figure 3 In the title of figure 3 there is one more space between the word risk-reducing I think they need to put perspectives in the conclusions                                                        Response) We changed figure title as following the reviewer comment and added perspective in the conclusion as: The intents for genetic test, RRM and RRSO were affected many factors. 

Reviewer 2 Report

This study analyses several factors that influence the willingness to undergo BRCA1/2 genetic testing and cancer risk-reducing surgery strategies in South Korean population. However, several clarifications can be useful to better understanding the survey responses. 

Line 112: Which information "about hereditary breast and ovarian cancer risk and BRCA1/2 testing" have been provided in detail? Have they been informed about percentage of increased cancer risk compared to general population, as well as the possibility of non-informative results from testing? And what about the possible side effects of cancer risk-reducing surgery strategies? 

Line 261: regarding "financial concerns". How much does genetic test cost in South Korea compared to a mean monthly salary? Was this information furnished to survey respondents? Are the survey respondent warranted about the cost of cancer therapies? In which percentage does the National Health System in South Korea cover costs of genetic testing, cancer therapies and risk-reducing surgeries? I believe these information are essential to understand the answers of low-income people.

Line 270: Is the "next-generation sequencing panel-based test" a multigene panel test or limited to BRCA1/2 analysis? In the first scenario, are the genes analysed only high-penetrant ones or also associated with intermediate risk of developing cancer? Was this information furnished to survey respondents?

Finally, the tables header have to be adjusted ("N=" does not appear near the number but on the line above).

Author Response

Thank you for informative comments. We changed manuscript according to reviewer's comment. 

Line 112: Which information "about hereditary breast and ovarian cancer risk and BRCA1/2 testing" have been provided in detail? Have they been informed about percentage of increased cancer risk compared to general population, as well as the possibility of non-informative results from testing? And what about the possible side effects of cancer risk-reducing surgery strategies? 

Response) When we first took the answer for HBOC risk and testing, we only explained simple knowledge the risk compared control vs. BRCA positive patients. We did not include possibility of non-informative results from test and possible side effects of cancer risk-reducing surgery. For this point, we added in the methods.

Line 261: regarding "financial concerns". How much does genetic test cost in South Korea compared to a mean monthly salary? Was this information furnished to survey respondents? Are the survey respondent warranted about the cost of cancer therapies? In which percentage does the National Health System in South Korea cover costs of genetic testing, cancer therapies and risk-reducing surgeries? I believe these information are essential to understand the answers of low-income people.

Response) Average household income was around $4,000 and the cost for BRCA1/2 was around $1,600 as total. However, cancer patients will pay $80 (5% of total cost) BRCA1/2 test in reimbursement system of Korea. When we asked for the genetic testing intention, we did not mention exact cost for tests and cancer therapies. For risk-reducing surgery, patient co-payment percentage is 5% for all costs. Although we did not fully asked for respondents in this view, we will include as the reviewer’s comment in next study.

Line 270: Is the "next-generation sequencing panel-based test" a multigene panel test or limited to BRCA1/2 analysis? In the first scenario, are the genes analysed only high-penetrant ones or also associated with intermediate risk of developing cancer? Was this information furnished to survey respondents?

Response) Next generation sequencing panel means multi-gene tests. When we did survey, we only included BRCA1/2 tests in the scenario and this include exact case scenario which did not include all indication for high or intermediate risk patients.

Finally, the tables header have to be adjusted ("N=" does not appear near the number but on the line above).

Response) We changed Table headers.

Reviewer 3 Report

The authors present a very large survey of the willingness to undergo BRCA1/2 genetic testing, risk-reducing salpingo-oophorectomy (RRSO), or risk-reducing mastectomy (RRM) among the general public, cancer patients, and healthcare professionals in South Korea. The study is in general well written and conducted. There are some concerns about what exactly respondents were told as the scenarios 1 and 2 in the supplementary give minimal information for instance not discussing contralateral breast cancer risk specifically nor any mention of ovarian cancer risk. A survey of the public talking about ‘prophylactic bilateral salpingo-oophorectomy’ would not be widely understood in English speaking countries. Overall this is still a worthwhile paper and the authors acknowledge the difference between intent and actual uptake.

Specific points

  1. ‘Th(e)s(e) results highlight(ed) the requirement of developing targeted educational materials and counseling strategies for facilitating informed decision-making’-These results highlight
  2. ‘Further considerations are required if a pathogenic variant is noted, which involve decisions regarding whether to elect (for) risk management options such as prophylactic surgery and share the test results with family members.’ -Most authorities prefer ‘risk reducing’ to ‘prophylactic’
  3. ‘Survey data were obtained from November 2016 to February 2017’ -why so much time elapsed to publish this?
  4. Scenario 2; ‘The test result of Mrs. Park revealed the presence of a BRCA mutation, which indicated that she was at high risk of developing breast cancer; she was advised to undergo a regular screening test or counseling with a doctor regarding undergoing preventive prophylactic salpingo-oophorectomy or prophylactic mastectomy.’ -Shouldn’t this be ‘high-risk’ of breast cancer in the untreated breast as she already has breast cancer. Also what is the ’direct translation for Korean. Do all women really understand what prophylactic bilateral salpingo-oophorectomy is?
  5. ‘Factors contributing to RRSO intent included “being at high risk of ovarian cancer” (72%), “commonness of ovarian cancer” (21%), and “family history of ovarian cancer” (6%).’ -The scenario does not mention ovarian cancer, how are people to know BRCA is associated if not already educated? FH of ovarian also seems very high for a cancer that only affects 1.5-2% in their lifetime.
  6. ‘Disclosure of genetic information is influenced by social, cultural, religious, and familial factors [9] and differs between Asian and Caucasian populations [14]; however, few studies have investigated these factors in Asia.’ There are studies from Malaysia you appear to have missed
  7. Please provide all information in supplementary provided to participants to make choices
  8. The authors do not note the trend for male general public and patients to have lower rates of acceptance of BRCA testing than women but male clinicians and researchers higher. This needs discussion and a statistical test

Author Response

 Thank you for all comments from reviewer. We explained the points and changed the manuscript.  

Specific points

  1. ‘Th(e)s(e) results highlight(ed) the requirement of developing targeted educational materials and counseling strategies for facilitating informed decision-making’-These results highlight

Response) Thank you for highlighting our conclusions.

2. ‘Further considerations are required if a pathogenic variant is noted, which involve decisions regarding whether to elect (for) risk management options such as prophylactic surgery and share the test results with family members.’ -Most authorities prefer ‘risk reducing’ to ‘prophylactic’

Response) There was an argument in Korea for prophylactic vs. risk reducing in committee of breast and ovarian cancer specialist. Then they prefer to use the word ‘risk reducing’ to prevent misunderstand as prophylactic of 100% prevention of cancer.

3. ‘Survey data were obtained from November 2016 to February 2017’ -why so much time elapsed to publish this?

Response) We tried this paper in several journals which took time to response and to the decision.

4. Scenario 2; ‘The test result of Mrs. Park revealed the presence of a BRCA mutation, which indicated that she was at high risk of developing breast cancer; she was advised to undergo a regular screening test or counseling with a doctor regarding undergoing preventive prophylactic salpingo-oophorectomy or prophylactic mastectomy.’ -Shouldn’t this be ‘high-risk’ of breast cancer in the untreated breast as she already has breast cancer. Also what is the ’direct translation for Korean. Do all women really understand what prophylactic bilateral salpingo-oophorectomy is?

Response) Even though the patient took breast cancer treatment, she is at high risk if she has mutation for BRCA1/2 genes. And prophylactic bilateral salphingo-oophorectomy was explained to remove ovary and salphix which mean it is not possible to give a birth and also it is quite well understood in the general population and patients after Angelina Jolie news.

5.‘Factors contributing to RRSO intent included “being at high risk of ovarian cancer” (72%), “commonness of ovarian cancer” (21%), and “family history of ovarian cancer” (6%).’ -The scenario does not mention ovarian cancer, how are people to know BRCA is associated if not already educated? FH of ovarian also seems very high for a cancer that only affects 1.5-2% in their lifetime.

Response) For high percentage of ovarian cancer, we included same number of ovarian cancer patients with other common cancer patients in this survey. Although we did not educate the respondents for ovarian cancer risk specifically, they understood it.

6.‘Disclosure of genetic information is influenced by social, cultural, religious, and familial factors [9] and differs between Asian and Caucasian populations [14]; however, few studies have investigated these factors in Asia.’ There are studies from Malaysia you appear to have missed

Response) We found recent paper of Malaysia and included in the discussion.

7. Please provide all information in supplementary provided to participants to make choices

Response) We added scenario in the supplementary materials.

8. The authors do not note the trend for male general public and patients to have lower rates of acceptance of BRCA testing than women but male clinicians and researchers higher. This needs discussion and a statistical test

Response) Thank you for comment. We added male general population showed difference for acceptance of test compared to clinicians in the results.